## Research Article

mental health; community-based initiatives; global mental health; group interventions; low income countries

**Corresponding author:**
Thandi Davies;
Email: thandi.davies@gmail.com

# Implementation outcomes of the waves for change community-based task-shared prevention intervention for adolescent mental health in South Africa

Thandi Davies[1] , Jamie Marshall[2], Nicola van der Merwe[3], Paula Yarrow[3], Tim Conibear[3] and Crick Lund[1,4]

[1]Alan J Flisher Centre for Public Mental Health, Department of Psychiatry and Mental Health, University of Cape Town Cape Town, South Africa; [2]School of Applied Sciences, Edinburgh Napier University, Edinburgh, UK; [3]Waves for Change, Cape Town, South Africa and [4]Centre for Global Mental Health, Health Service and Population Research Department, Institute of Psychiatry, Psychology and Neuroscience, King's College London, London, UK

## Abstract

Adolescents face increased vulnerability to mental health conditions, particularly when exposed to multidimensional poverty and trauma and pronounced treatment gaps. Waves for Change, a Sport for Development (SfD) intervention, employs task-sharing through its '5-Pillar Method' to build resilience and prevent mental health conditions among at-risk adolescents in South Africa. This study assessed the implementation outcomes of this Method using a mixed-methods design, incorporating interviews, focus groups, self-report questionnaires, document reviews and routine site assessments, with 69 stakeholders including staff, peer coaches, mental health-care providers, social workers, teachers, and adolescents. A thematic analysis revealed key facilitators to successful implementation, including: a year-long preventative approach, creation of safe spaces for learning self-regulation skills, employment of youth coaches from local communities, incorporation of fun, group-based physical activities, modelling and repetition of desired skills, provision of transport and meals, government partnerships, and consistent weekly training and supervision. Implementation challenges included coach capacity, due to their education levels and own trauma experiences, measurement of fidelity to the Method and of adolescents' emotional experiences, and some concerns around ocean safety. These findings provide valuable insights for implementing community-based SfD interventions for adolescents facing adversity, and contributes towards global evidence supporting task-shared mental health approaches in LMICs.

## Impact statement

Adolescence is a crucial period of development, and the presence of multidimensional poverty produces an environment of toxic stress that intensifies the risk of developing mental health conditions. However, there is a large treatment gap in the provision of adolescent mental health services in low- and middle-income countries (LMICs). Using task sharing, sport for development interventions can play a crucial role in promoting resilience and preventing the development of mental health conditions during adolescence. Through a mixed-methods design and interviews with 69 stakeholders, this study evaluated the implementation outcomes of the 5-Pillar Method used by the Waves for Change sport for development organisation as a task-shared prevention intervention for adolescent mental health conditions in South Africa. The method aims to introduce and strengthen protective factors for children experiencing toxic stress, and improve their ability to self-regulate while living in adverse conditions. Findings demonstrated that the 5-Pillar Method is a feasible, acceptable, appropriate and sustainable method for delivering a social and emotional skills programme to adolescents exposed to toxic stress and social and economic adversity. This is facilitated through having a year-long programme with a trauma-informed preventative approach to mental health, the creation of safe physical and emotional spaces for optimal learning of self-regulation skills, using youth coaches from the community, including fun group-based physical activities, modelling and repetition of desired skills, provision of transport and a meal and weekly training and supervision of coaches throughout the programme. This study provides evidence of the implementation outcomes of a community-based task-shared Sport for Development intervention for at-risk adolescents and may be applied to similar situations in other LMICs.

## Introduction

Adolescence is a crucial period of development, where the foundations of future health are laid down (Blakemore, 2012; WHO, 2017). Almost one-third of adolescents in low- and middle-income countries (LMICs), including South Africa, live in households that face monetary, educational and service infrastructure deprivations and are multidimensionally poor (Alkire et al., 2019), with major challenges related to substance abuse by parents and youth (WHO, 2018). A national study found that 42% of children in South Africa had experienced maltreatment, 33% had experienced sexual or physical abuse and 20% had experienced neglect (Artz et al., 2018).

The repetitive trauma and adversity related to these deprivations can produce a toxic stress response in adolescents, which refers to "maladaptive coping skills, poor stress management, unhealthy lifestyles, mental illness and physical disease" (Centre on the Developing Child, 2014; Franke, 2014). This can lead to long-term adverse psychological and physical health outcomes (Beranbaum et al., 2023). Childhood trauma and poverty have also been found to increase the risk for mental health conditions, such as depression, anxiety and post-traumatic stress disorder (Varese et al., 2012; Mauritz et al., 2013; Haushofer and Fehr, 2014; Palacios-Barrios and Hanson, 2019; WHO, 2020).

With adolescents comprising 19% of the total population of South Africa (Statistics South Africa, 2018), investment in evidence-based interventions that respond to contexts of trauma and toxic stress is vital (WHO, 2021; Alckmin-Carvalho et al., 2022). However, there is a major 'treatment gap' for adolescents requiring public mental health services in South Africa (Docrat et al., 2019). Fewer than 10% of children and adolescents who need a mental health service receive it, with contributing factors including a lack of facilities, trained professionals and culturally appropriate services (Tomlinson et al., 2022).

A 'task-shared' approach towards mental health services is therefore necessary, whereby community health workers or peer educators take on responsibilities previously delivered by specialists in providing psychosocial support to at-risk adolescents under the supervision of more specialised providers (Fairburn and Patel, 2014; WHO, 2021). This approach can increase coverage of basic care and prevention of serious mental disorders through early identification and support. This is in line with the World Health Organisation's (WHO) Mental Health Action Plan (2013–2030) and the recent South African Mental Health Policy Framework and Strategic Plan (2023–2030). These plans recommend scaling up community health services and engaging non-governmental organisations (NGOs) to strengthen an intersectoral approach to mental health (WHO, 2013; RSA Department of Health, 2023).

Physical activity and sports have been identified as active ingredients in task-shared psychosocial interventions for adolescents (Teychenne et al., 2020; Pote, 2021; WHO, 2021). The use of sport can be an important 'draw' factor into psychosocial interventions, through experiences of fun, friendships and structured play. Using sports coaches as peer educators can make task-shared sports programmes a viable vehicle to bridge the prevention gap for adolescent mental health conditions (Holt et al., 2017).

The combination of physical activities with social and emotional skill interventions, known as Sport for Development (SfD) (Lauwerier et al., 2020), can lead to positive mental health outcomes (Sherry and O'May, 2013; Marshall et al., 2021; Boelens et al., 2022). An exception to this was a football intervention in Uganda, where a negative effect on the mental health of participants was attributed to the focus on competitive sport without an intentional focus on mental health components (Richards et al., 2014).

Surfing is gaining recognition as an SfD tool for improving psychosocial health outcomes for adolescents (Benninger et al., 2020). Studies have found surf therapy to have positive impacts on mental health and social connections, social skills, well-being, self-management, and attitudes towards school (Godfrey et al., 2015; Hignett et al., 2018; Marshall et al., 2020; Marshall et al., 2021).

'Waves for Change' (W4C) is a South African NGO that uses surfing to improve the psychosocial well-being of adolescents. It was established in 2011 as a surfing club for youth in Cape Town, in the recognition that surfing was a novel way to engage youth. Realising that local social services were under-resourced, they conducted participatory research with children to provide child-led recommendations for an integrated surfing and psychosocial programme (Benninger and Savahl, 2016). This research found that "feelings of safety, social connectedness and children's spaces" (p. 10) were central for children to construct and assign meaning to the 'self'. The study recommended that the programme include activities to improve "children's self-concept, including the construction of safe spaces for children to play, learn and form meaningful relationships" (p. 1).

A programme was developed over the following years, including further research that identified mechanisms that were foundational towards achieving outcomes of well-being, social connectedness and improved mental health (Marshall et al., 2020). This was created into the organisation's '5-Pillar Method', which has the following components:

1) Consistent and caring adults and positive peers.
2) Access to safe spaces (physical and emotional).
3) Fun and challenging new tasks (e.g., surfing or other group-based physical activities).
4) Learning social and emotional skills.
5) Connections to new opportunities and services.

Given the growth of W4C since 2011, and the role it may play in filling the mental health treatment gap for at-risk adolescents, this study aimed to evaluate the implementation outcomes of the W4C 5-Pillar Method as a task-shared prevention intervention for adolescent mental health in South Africa. This will provide a platform from which to assess the effectiveness of the W4C programme, with the broader aim of scaling up community services and reducing the burden on specialist services (RSA Department of Health, 2023).

The evaluation of key implementation outcomes assesses how well implementation has occurred, focusing on the 'practicality' of interventions in real-world contexts (Proctor et al., 2011). This informs intervention outcomes in future effectiveness research, and ultimately optimises strategies for the scale-up of psychosocial interventions (Wagenaar et al., 2020; Seward et al., 2021). Evaluations with a vision for future scale up can also contribute towards the availability of psychosocial interventions for others to use (Rose et al., 2025).

Proctor et al. (2011) published a taxonomy of eight implementation outcomes for the evaluation of interventions, including acceptability, adoption, appropriateness, feasibility, fidelity, implementation cost, penetration and sustainability. This is summarised in Table 2 in the methods section. With the exception of cost, this taxonomy was used to guide an evaluation of the implementation outcomes of the 5-Pillar model.

## Methods

Methods and findings are reported using the COnsolidated criteria for REporting Qualitative (COREQ) studies 32-item checklist as a guideline (Tong et al., 2007). Responses to the COREQ checklist are provided in Supplementary Document 6.

### W4C programme

The W4C programme uses the 5-Pillar Method to introduce protective factors to adolescents exposed to toxic stress and improve their ability to self-regulate while living in adverse conditions. The Method was primarily designed to optimise the natural benefits of group-based sport, recreation and play to move distressed pre-symptomatic adolescents (10–16 years) towards wellness (Marshall et al., 2020).

Adolescents attend one session per week for 10 months across one calendar year, following which they graduate to a 'surf club' on Saturdays. Each 2-hour session aims to provide adolescents with a predictable routine in a safe space to support nervous system regulation. Self-regulation tools are taught through fun games and catchy phrases, such as doing a 'Take-5' (five mindful breaths) or using 'snap claps' to demonstrate appreciation and praise of others. Each self-regulation game includes key messages and clear steps to help adolescents reach session objectives and learn through experience. The curriculum is run for 5 months, and then repeated to entrench learnings.

Coaches aged 18–25 are employed in the programme and come from the same communities as the adolescents. They receive an initial 5-day 'grounding' training, followed by weekly group training and supervision throughout the year. They are trained in empathic coaching skills (e.g., active listening, giving praise, managing and containing trauma disclosures), how to make sessions fun and playful, first aid, child protection, surfing, and water skills, with the 5-Pillar Method used as a broad framework for training. Coaches then embed the 5 Pillars in sessions through a routine of 'energisers', a check-in, a self-regulation-based activity from a scaffolded curriculum, surfing/play time, and a check-out. An implementation manual is used, which includes simple language and basic implementation instructions. Implementation relies heavily on pre-session training in conducting sessions and modelling appropriate behaviours. Further details of the training programme and curriculum can be found in Supplementary Document 1.

### Study design

The study employed a mixed-methods design, including exploratory data collected through individual semi-structured interviews and focus group discussions (FGDs), a self-report questionnaire, document reviews of internal and public reports, and routine site assessment data. The lead researcher developed the topic guides and sent these to fellow authors for comment. These guides followed the same format but with customised questions for different stakeholder groups.

### Setting

The study was conducted in the five ocean-side sites where the programme is offered in South Africa. These are Muizenberg, Khayelitsha, and Hout Bay in Cape Town, Western Cape province; and East London and Gqeberha in the Eastern Cape province. The adolescents are aged between 10 and 16 years old, and they are referred to the programme from the schools and mental health service providers in the surrounding communities.

### Sample and recruitment

Adult participants were purposively selected for their roles with or in W4C. This included W4C trainers, curriculum developers, a Child Protection Officer, and coaches with experience ranging from a few months to 4 years; mental health providers who worked in the public health sector and collaborate with W4C; and referring partners such as teachers and social workers who connect adolescents to the programme.

The adolescent participants were from six groups who were attending the weekly W4C programme at the three programme sites in Cape Town (two groups per site) in 2023. Adolescents and coaches from the East London and Gqeberha sites were not included as these focus groups would have had to be held online and this was deemed unfeasible by the W4C site managers.

A de-identified list of all adolescents attending the three programme sites was obtained from W4C and initially stratified by site and gender. Following this, the adolescents were randomly selected using a random number generation form (http://www.jerrydallal. com/random/randomize.htm). Eight adolescents were selected for each group, but not all of them arrived on the designated day, resulting in a total of 39 adolescent participants (see Table 1 for data on the adolescent participants).

**Table 1.** Adolescent focus group participant characteristics

| Demographic | |
| --- | --- |
| Age (10–15) | Sample size (n = 39) |
| 10 | 2 |
| 11 | 10 |
| 12 | 18 |
| 13 | 4 |
| 14 | 2 |
| 15 | 1 |
| Gender | |
| Female | 19 |
| Male | 20 |
| Site | |
| Khayelitsha | 14 |
| Hout Bay | 12 |
| Muizenberg | 13 |
| Attendance (17–100%) | |
| 80–100% | 15 |
| 60–79% | 16 |
| 17–59% | 8 |

### Research process

Individual interviews were conducted with five W4C staff members and five mental health providers. Written questionnaires were sent to seven teachers and social workers. Two FGDs were held with

13 W4C coaches, and five FGDs were held with 39 adolescents. All interviews were conducted by the lead researcher (TD), who was independent of the organisation. Coaches and adolescents were assured that their feedback would be confidential and that what they shared would not impact their participation or employment in any way. Social desirability bias was minimised through TD not being known to the participants (apart from two staff members). Interviews were conducted in English, which was participants' first or second language. Individual interviews took between 30 and 60 minutes, and FGDs lasted between 60 and 90 minutes. Interviews were transcribed verbatim by a professional transcriber who signed a confidentiality agreement. Supplementary Document 2 contains further details on these FGDs.

Written informed consent was obtained from all participants and the caregivers of the adolescents, together with assent from the adolescents. Ethical approval for the study was obtained from the University of Cape Town Health Sciences Faculty Human Research Ethics Committee, reference number 790/2022. The research was performed in accordance with the Declaration of Helsinki.

### Analysis

#### Qualitative data

FGDs and interviews were audio-recorded, transcribed and imported into NVivo 12 for analysis, along with the self-report questionnaire responses. Qualitative data analysis was conducted using a 'theoretical thematic analysis' (Braun and Clarke, 2006). This process allowed themes to be explored in relation to the research questions (a priori themes), and for new themes to emerge from the data. Following thorough reading of the data, initial primary codes were created. These codes differed from the themes, which were broader and included both a priori and new themes that emerged from the analysis.

The a priori themes were defined using a taxonomy of implementation outcomes published by Proctor et al. (2011) and included appropriateness, adoption, acceptability, feasibility, fidelity, penetration and sustainability. The outcome 'cost' was not included as we did not have the assessment tools required to conduct a robust estimate of the cost of the programme. Data were primarily coded by TD. To assess the consistency and accuracy of coding, codes and themes were presented to the non-W4C authors for feedback. Table 2 provides definitions and terms of implementation outcomes identified by Proctor et al. (2011), which were used as a priori themes in the data analysis.

#### Document reviews

W4C supplied historical documents and annual reports for background and process information. These were used to report on elements such as how adolescents are referred to the programme, which schools are involved, and how many adolescents attend every year.

#### Routine site assessment data

W4C collects routine internal fidelity data to assess the implementation of routine activities and core actions using a 'Site assessment tool'. Data are collected twice weekly by a wide range of W4C staff members (e.g., national director, M&E officer, site managers, training managers, training officers, senior coaches). The 50 site assessment questions are supplied in Supplementary Document 3. For this study, the data from these assessments over 3 years (2021–2023), representing 21% of all sessions, were examined using descriptive analysis to assess fidelity to the 5-Pillar Method.

#### Reflexivity

It is important to note that three authors are involved in the broad-level operation of the programme, which leaves room for potential biases. To mitigate this, an independent researcher (TD) was appointed to conduct the study. This researcher conducted all interviews, where she probed for both positive and negative feedback. She also conducted the data analysis and interpretation, and then approached the two other independent authors to offer insight. The W4C authors were, therefore, not involved in participant selection, interviewing, data coding or interpretation in this study.

The primary author (TD) also included reflexive practise elements throughout the research, such as reflexive interviewing, noting how tone and positionality may influence responses, memo writing, taking field notes to note interactions and emotions during fieldwork, using audit trails to document any decisions, and peer debriefing to receive feedback to challenge any assumptions. Her

**Table 2.** Implementation outcome definitions and terms, identified by Proctor et al. (2011)

| Implementation outcome | Definition | Terms identified from other literature |
|---|---|---|
| Appropriateness | The perceived fit, relevance or compatibility of the innovation or evidence-based practice for a given practice setting, provider or consumer; and/or perceived fit of the innovation to address a particular issue or problem. | Perceived fit; relevance; compatibility; suitability; usefulness and practicability |
| Adoption | The intention, initial decision or action to try or employ an innovation or evidence-based practice. | Uptake; utilisation; initial implementation and intention to try |
| Acceptability | The perception among implementation stakeholders that a given treatment, service, practice or innovation is agreeable, palatable or satisfactory. | Satisfaction with various aspects of the innovation (e.g., content, complexity, comfort, delivery and credibility) |
| Feasibility | Feasibility is defined as the extent to which a new treatment, or an innovation, can be successfully used or carried out within a given agency or setting. | Actual fit or utility; suitability for everyday use and practicability |
| Fidelity | The degree to which an intervention was implemented as it was prescribed in the original protocol or as it was intended by the program developers. | Delivered as intended; adherence; integrity and quality of program delivery |
| Penetration | The integration of a practice within a service setting and its subsystems. | Level of institutionalisation; spread and service access |
| Sustainability | The extent to which a newly implemented treatment is maintained or institutionalised within a service setting's ongoing, stable operations. | Maintenance; continuation; durability; incorporation; integration; institutionalisation; sustained use and routinisation |

background is a privileged one as a white, English-speaking, middle-class individual. This could have imposed power dynamics on some of the participants in the study. However, she has extensive individual and group interviewing and facilitation experience, and is also trained as a Registered Counsellor with the Health Professions Council of South Africa.

## Results

Table 3 provides a summary of the participants, methods, sample sizes and anonymisation labels given to participants. Seven implementation outcomes of the 5-Pillar Method are reported below.

**Table 3.** Participant sample sizes, data collection methods and participant labels

| Participants | Method | Sample size (n = 69) | Participant labelling |
|---|---|---|---|
| W4C staff members (programme developers and trainers) | Individual semi-structured interviews | 5 | S1–S5 |
| W4C Coaches | Focus groups (2) | 13 (6, 7) | C1–C13 |
| Mental healthcare providers (psychologists, counsellors and social workers) | Individual semi-structured interviews | 5 | HC1–HC5 |
| Partner stakeholders (teachers or healthcare providers) | Open-ended questionnaires | 7 | SK1–SK7 |
| Adolescents attending the programme (aged 12–15) | Focus groups (5) | 39 (13, 5, 7, 7, 7) | AD1–AD39 |

### Appropriateness

All stakeholders mentioned a wide variety of challenges that adolescents in the programme face. These included food insecurity, poverty, absent parents, lack of praise and attention, having no one to talk to, abuse, unsafe play areas, unsafe communities, abandonment, poor life skills, no role models and gangsterism.

Responding to these needs improved the perceived fit and compatibility of the Method for adolescents, and participants identified elements of the 5-Pillar Method that were appropriate, suitable and useful to the adolescents in the context of the psychosocial and economic challenges they face. For example, being able to be in the ocean, catch waves, play games and feel physically safe provided a feeling of respite for children who have been exposed to trauma and who do not have any other physical spaces that are safe to play in, demonstrating the suitability of having access to safe spaces. A coach explained this: "*Seeing them in the water, they're having fun, they are not having to think of, 'what am I going to face when I get home, what am I going to eat today?' They don't really think of the outside world when they're in the water*" *(C3)* (also see Supplementary Document 5, Q1&2).

The organisation's approach to teaching and learning social and emotional skills through creating safe physical and emotional spaces, and fun tasks using sport, role-plays and repetition (Pillars 2–4) stems from recommendations from previous research (Benninger and Savahl, 2016; Marshall et al., 2020). A more recent study has also impacted appropriateness after measuring heart-rate variability with 83 W4C adolescents (Beranbaum et al., 2023). A staff member explained that "*this study informed our learnings around dosage and how long it was taking our participants to settle into the space*," and as a result have extended the curriculum to add a first phase of 8 weeks that "*focuses entirely on coach behaviours to create safe spaces and build strong connections*" (S1) (also see Supplementary Document 5, Q3). She further described that the curriculum is trauma-sensitive and based on Humanistic therapy, Acceptance and Commitment Therapy principles, behaviour activation, goal setting, emotional regulation, breathing and meditation techniques and positive youth development (S1).

Coaches and staff related that the 5-Pillar Method is a prevention intervention, which they believed is appropriate for the adolescent group that they serve. A staff member explained that the organisation is clear about its scope of practice, targeting at-risk but not diagnosed adolescents, to ensure usefulness and practicability for a large number of adolescents in need: "*We encourage our coaches to not even necessarily advise. It really is around creating safe, relaxing, fun spaces*" (S1).

Coaches explained that creating a relaxed, fun and non-pressurised environment was more appropriate for adolescents than traditional talk therapy with specialists: "*It's kids, remember. You can't expect the kid to sit with you and then start telling you stuff and all that. At least you need to give them something fun. And try to put yourself in their shoes*" *(C9)*. Supplementary Documents 4 and 5 (Q10) provide further examples of coaches intentionally creating safe spaces for adolescents.

If coaches identify that a child is struggling, or discloses abuse, they immediately refer them to the delegated Child Protection Officer at the site, and/or their managers. A coach explained that if this happens, "*I have to like, speak to my site manager, like [Mr X], so that he can refer the child to someone at the job or someone who is a manager. Then they take it from there*" (C12).

Recruiting appropriate coaches to meet requirements for both professionalism and organisational values was mentioned as a challenge. A staff member explained that they "*try and do values-based recruitment but it's tricky because the coach is the key driver of impact our programme, so their professionalism is also important*" (S1). The psychologists also stated that coaches should ideally have attributes that are appropriate for working in mental health, such as having an inclination towards mental health, and being old enough to provide good life examples and serve as good role models for children.

### Adoption

Coaches and adolescents provided frequent accounts of the coaches' utilisation and adoption of the behaviours and teachings required from the Method for 4 of the 5 Pillars, and moderately, for the fifth Pillar. Other stakeholders also reported uptake of the behaviours by adolescents (see Supplementary Document 4 for data demonstrating this). This adoption is facilitated by many of the factors mentioned above, and also through the organisation's approach to learning, through repetition and "*encouraging awareness within the participants about how a behaviour made them think*

and feel, and then planning together how they can use those in their lives" (S1) (also see Supplementary Document 5, Q4).

Adoption of the Method by coaches as non-specialist providers was made viable by the format of training and supervision that the organisation uses. Coaches related that the initial 5-day grounding training helped them to understand the key concepts underlying the 5 Pillars, such as how to create a safe space for adolescents. Following this, group training and supervision are held weekly, and throughout the year, which improves both adoption and fidelity. New modules are introduced to coaches each week before they conduct them with the children, ensuring that activities are always "fresh in their minds" (S1). Coaches are also asked to practice the skills and reflect on how they would use them in their own lives. Through this, they obtain a deeper understanding of the exercises and reasons for doing them (S3). Three past coaches have also been promoted to training officers, which helps to clarify expectations and demonstrate desired behaviours.

Another way in which adoption of the Method is improved is the active creation of an organisational culture whereby all staff adopt and enact the behaviours and values of the Method, so that coaches, in turn, model these for the adolescents. This was described as such: "*We ensure that we model the behaviours mentioned in our five pillars. We make it part of our training every week so that it becomes something like muscle memory. So it becomes a habit for [the coaches] to also start modelling those positive behaviours when they are with the kids*" (S4). "*So, it's your behaviours that will create a culture that will create a safe space, which boosts well-being*" (S1).

Staff members explained that because coaches come from the same background and educational context as the children, "*they themselves sometimes struggle to self-regulate*" (S4), which can prove a challenge to the implementation of the Method. Weekly training, therefore, always includes the core skills for coaches and adolescents, such as breathing regulation, meditation, self-regulation and the creation of a safe space.

Another challenge to the adoption of the Method was that there appeared to be an unclear understanding or directive regarding the fifth Pillar. Coaches demonstrated knowledge of referral pathways to the Child Protection Officers for mental health issues, but there was little evidence of 'connections to other opportunities'. A staff member said that the coaches "*don't always have a fully common understanding of what it involves*" (S5), which was visible through the focus groups with the coaches themselves. However, access to one follow-up service was "*the opportunity to be part of surf club*" (S4).

### Acceptability

Participants perceived the 5-Pillar Method to be satisfactory and relevant, primarily because it uses coaches who are young (aged 18–25) and from the same communities as the adolescents. According to the participants, this enables coaches to have a strong understanding of the social, economic, educational and cultural context of the adolescents (SK1, SK2, HC1), which cannot be taught through didactic training, thus improving acceptability. Coaches were described as "*relatable*" (SK2), and "*understanding of what children need in a space, coming from the same space and experiencing the same things as the children*" (S2). A coach explained this: "*we know the communities, the background, where they live in the households… We know how hard it is and we went through that*

already. We've been to school already so we will be able to help them and encourage them that there is an outside world out of where they're living in. [That] there is better, and they can get better" (C5).

Local youth coaches also serve as role models for the adolescents, where parental and social role models are scarce. School teachers explained that: "*our learners don't have mentors at school, and they do not have any role models. [So they] tend to look up to the coaches*" (SK4).

Practically, all adolescent participants confirmed that it was "easy" for them to attend the programme every week. They additionally reported satisfaction with the techniques they were being taught and reported learning how to: 'celebrate others, do the 'Take 5′, be yourself, take care of people, respect others, check in with others and trust others' (AD1–39). They also described elements of the programme that they enjoyed, which included: swimming, learning water skills, surfing, being in the ocean and on the beach, receiving food, learning about the ocean, being with the coaches, doing breathing exercises, having the opportunity to relax, having fun, getting snap claps (appreciation from coaches and fellow adolescents), being part of the 'banana' culture (see Supplementary Document 5, Q5), and meeting children from different schools (AD1–39). A coach related that one of the benefits of being in the ocean for adolescents was the chance to "*have time to think and get in touch with your inner self*" (C7) – something that was perceived as near-impossible in a township context.

Only a few barriers to acceptability for the adolescents were mentioned. These involved concerns about physical safety, such as being afraid of big/rough waves, being worried about drowning or worried about falling off the board into the water. One group felt that they had been unfairly blamed as a group for a few specific children's bad behaviour. Lesser-mentioned concerns included arguments with fellow adolescents, the occasional possession being stolen or bullying from adolescents from other schools.

Teachers and care workers mentioned that the lack of involvement of parents in the programme could be a barrier to intervention effectiveness. One explained that "*the problems start mainly in their homes,*" and so she would suggest "*involving parents more and having sessions with them as well. Because as they are also vulnerable, because of socioeconomic issues, poverty, crime and violence. They tend to expose the children and they don't know how these impact their children's lives*" (SK6). These stakeholders suggested adding skills programmes for parents to address this. A staff member, however, described the remit of their programme, and said, "*we've recognised that we can't change the communities, but we* can *change the participants*" (S1).

### Feasibility

Feedback from stakeholders and adolescents was that a year-long programme was feasible for them, although many adolescents said that that they would like the programme to go on beyond a single year. A few adolescents expressed difficulties attending when parents were worried about safety in the ocean, or when they had homework, or had to clean the house, or when there were "*home issues – like if somebody does something and you have to handle it, it's not easy*" (AD15).

Adolescents and coaches are all provided with transport to and from the programme, which encourages consistent attendance from both coaches and adolescents, and contributes significantly

to the feasibility of adolescents being able to attend. The programme is made *"easily accessible to adolescents through being free"* (SK2). Adolescents are also provided with a meal after every session.

The psychologists in the study endorsed the 5-Pillar Method but had concerns about the coaches' own levels of trauma, and their ability to contain and counsel traumatised adolescents. One described this, saying, *"I think that sometimes the young people that we train and put in these positions are not necessarily equipped to deal with it. For instance, if a mentor is paired with a young adolescent who is raped, and that mentor themselves has been raped, you are expecting that untrained person to contain that child, but who is containing the mentor? What happens when you are triggered? So, they do have the potential to do this but they need containment and counselling themselves"* (HC4).

A coach confirmed that *"sometimes you do get triggered by the stories that you hear but as a coach you have to be strong for the participants"* (C10). However, they further explained that they have support systems in place if needed, through the provision of a psychologist once a week at every site *"that we get to share, like, reflect on the week"* (C2). As described, they are also trained to refer adolescents who disclose traumatic incidents or abuse directly to the Child Protection Officer at the site.

Coaches also mentioned that they use the self-regulation techniques themselves, demonstrating suitability and feasibility of these skills for youth. One coach explained: *"If I get triggered, I just try to calm myself down, do a meditation, and do a 'take five' [mindfulness activity]"* (C2).

## Fidelity

Fidelity to implementation of regular activities is measured by the organisation using a 50-item 'Site assessment tool' (see Supplementary Document 3). A range of staff conduct these assessments, including coaches, which encourages fidelity to these factors and is also used in supervision sessions. In this assessment, eight items relate to implementing elements of the 5-Pillar Method itself. Fidelity to these eight items was analysed using percentage distributions of 'Yes', 'No' and 'Only some', across the whole dataset. All assessments conducted over a 3-year period were included, totalling 857 assessments, which represent 21% of all sessions conducted. From these items, fidelity was high, ranging from 73% to 95%. Table 4 below provides percentages of engagement with the eight key questions from the site.

While the assessment measures 50 items relating to the implementation of the programme as a whole, it does not include less 'observable' elements such as the creation of a safe space and a 'caring culture' by coaches. This challenge was confirmed by a staff member who stated that it was difficult to measure 'softer outcomes' such as coach performance, whether the core messages were being communicated in each session, as well as adolescents' emotional or subjective experiences, *"such as whether the children are having fun"* (S1).

Using local young coaches also brings in the need for a nuanced balance between encouraging coach autonomy and independence in implementing the programme, and maintaining a standardised approach that is *"evidence-based and intentional"* (S1). A staff member related that *"we haven't always got the balance right… between making sure that they can adapt it and run with it, while at the same time, the core ingredients have to remain… so we've*

**Table 4.** Routine site assessment fidelity data (n = 857 sessions, over 3 years, representing 21% of all sessions conducted)

| Site assessment question | Yes | Only some | No | Data not collected |
|---|---|---|---|---|
| Children greeted enthusiastically | 88,8% | 1,9% | 3,1% | 6,2% |
| Coaches/children participated with energy in the energiser/ warm-up activity | 92,9% | 2,6% | 1,6% | 2,9% |
| Coaches sat in a safe circle and practiced calm breathing | 94,6% | 1,3% | 1,4% | 2,7% |
| Coaches did a check-in with last week 'Teachable Moment'? | 90,2% | 1,3% | 5,8% | 2,7% |
| Coaches asked children 'what' they learnt last week and 'how' and 'when' they used it? | 72,8% | 0,0% | 15,3% | 11,9% |
| Coaches explained the lesson/ behaviour for the week? | 93,3% | 1,7% | 2,3% | 2,7% |
| Coaches engaged in this week's lesson/ behaviour? | 94,5% | 1,0% | 1,7% | 2,7% |
| Coaches did a debrief summary/ check-in with children after the lesson? | 92,3% | 1,6% | 3,3% | 2,7% |

*sometimes left [the implementation] a bit too open and that is then actually not empowering and enabling for coaches to do their job… But at other times, we've been too prescriptive, which has not allowed for their local understanding and their own essential knowledge to come through… So, how we train it and encourage them in the underlying culture, we've had to learn"* (S1) (also see Supplementary Document 5, Q6).

## Penetration

W4C started as a surf club with 15 adolescents and 2 coaches in Masiphumelele, Cape Town, in 2011. In 2024, the organisation had 51 partner schools and 18 referring community health providers, including those from the Department of Health, across 5 sites, employing 43 coaches, and with 1,493 adolescents attending the programme. Most current partners have been involved with W4C for between 3 and 5 years. In 13 years, the organisation has reached a total of 14,618 adolescents (see Table 5).

In 2024, there were 52 government schools in the five sites. Given that the organisation partners with 51 of these schools, this demonstrates excellent school coverage and partnership in the programme sites. This has been aided by long-term partnerships with schools and health providers, buy-in from the government and the provision of checklists for teachers to more easily identify at-risk adolescents for referrals.

Penetration within the 51 schools is 1,493 adolescents out of approximately 51,644 students in total (2.9% coverage, see Table 6). Individual reach is limited by organisational capacity and access to safe and practical beaches. To address this and increase their penetration beyond surfing and the ocean, W4C partnered with the Department of Cultural Affairs and Sport in 2023 to integrate the 5-Pillar Method and teaching routine into their governmental after-school sports programmes. W4C is providing three-day baseline training and monthly top-up training to these sports coaches.

**Table 5.** Waves for Change penetration and reach since 2011

| Year | Number of new adolescents attending at least one session | Number of adolescents attending surf club | Total adolescent reach per annum | Number of referring partners and schools | Number of coaches employed | Of which new coaches |
|---|---|---|---|---|---|---|
| 2011 | 15 | N/A | 15 | 4 | 3 | 3 |
| 2012 | 30 | 15 | 45 | 4 | 2 | 0 |
| 2013 | 70 | 30 | 100 | 4 | 6 | 4 |
| 2014 | 80 | 70 | 150 | 4 | 10 | 6 |
| 2015 | 140 | 80 | 220 | 6 | 18 | 13 |
| 2016 | 160 | 140 | 300 | 6 | 19 | 15 |
| 2017 | 377 | 160 | 537 | 34 | 21 | 20 |
| 2018 | 730 | 161 | 891 | 49 | 33 | 22 |
| 2019 | 777 | 502 | 1,279 | 49 | 41 | 22 |
| 2020 | 1,500 | 502 | 1,500 | 49 | 51 | 22 |
| 2021 | 1,497 | 522 | 2,019 | 49 | 43 | 22 |
| 2022 | 1,434† | 874 | 2,308 | 52 | 43 | 22 |
| 2023 | 1,627‡ | 1,027 | 2,654 | 69 | 43 | 22 |
| 2024 | 1,493* | 1,107 | 2,600 | 69 | 43 | 22 |
| **Total** | **9,930** | **5,190** | **14,618** | **69** | **376** | **215** |

†Of these 1,434, 82 dropped out (5.7%).
‡Of these 1,627, 91 dropped out (5.6%).
*Of these 1,493, 149 dropped out (10%).

**Table 6.** Number of schools and children in the five participating programme sites

| Site | Number of Schools | Total number of children in these schools |
|---|---|---|
| Muizenberg | 11 | 16,171 |
| Khayelitsha | 12 | 13,813 |
| Hout Bay | 5 | 3,295 |
| East London | 12 | 10,063 |
| Gqeberha | 12 | 8,302 |
| Total population | 52 | 51,644 |
| W4C 2024 data | 51 | 1,493 |

### Sustainability

Being an NGO and not a for-profit company means that the organisation is reliant on external funding, which is a threat to sustainability. However, W4C has managed to obtain funding every year from Trusts, Foundations and private Philanthropists, with smaller amounts of funding also coming from individual givers. This enables the maintenance of a strong administrative team coordinating logistics, and a stable, long-term staff base. Relationships with provincial and local government also allow W4C to benefit from the use of city buildings without having to pay for rent.

Sustainability is also enhanced through well-established and stable relationships that W4C has with referral partners, such as social workers and mental health practitioners in the Department of Health. The organisation has been in partnership with the Western Cape Education Department since they started in 2011, through training teachers to identify and refer adolescents at risk of mental health disorders using a list of indicators, indicating greater, systemic adoption and sustainability.

### Discussion

The WHO recommends universal prevention and promotion interventions for younger populations in contexts of poverty (WHO, 2020), where the risk for toxic stress and mental disorders is increased, and for adolescents already experiencing emotional problems (WHO, 2021). In this study, stakeholders found that a preventative approach to mental health, through the creation of safe physical and emotional spaces, using fun tasks, and enabling a feeling of respite, was appropriate for responding to toxic living contexts. These factors support adolescents to more readily be able to learn coping and self-regulation skills (Marshall et al., 2020), which in turn have the potential to introduce protective factors that buffer the influence of poverty and trauma on mental health (Troy and Mauss, 2011; Palacios-Barrios and Hanson, 2019). It has also been found that sporting interventions that emphasise fun, provide an opportunity for individual mastery, and are non-competitive are most likely to benefit at-risk youth (Petitpas et al., 1999) and improve attendance (Holt et al., 2017).

The study found that the 5-Pillar Method was also appropriate for at-risk adolescents in South Africa due to its provision of context-relevant self-regulation skills, befitting the circumstances of toxic stress that the adolescents face. The organisation has adopted recommendations from previous research conducted with its participants (Benninger and Savahl, 2016; Beranbaum et al., 2023), and has developed a trauma-sensitive curriculum that is supported by other research (Livheim et al., 2015; Pote, 2021; Jenness et al., 2023).

The use of youth coaches from the same communities as the adolescents was seen as appropriate, acceptable and feasible to study participants, and has been found to improve acceptability in other task-shared interventions for adolescents and adults in LMICs (Verhey et al., 2020; Ceccarelli et al., 2023). This task-shared approach to address the mental health treatment gap aligns with recommendations from the WHO mental health Gap Action Programme, the WHO adolescent mental health toolkit and the South African Mental Health Policy Framework (WHO, 2016, 2021; RSA Department of Health, 2023). It has now been tested in a wide number of psychosocial interventions for youth in LMICs (Rose et al., 2025).

One challenge in conducting interventions with fidelity with youth coaches was finding the balance between giving coaches autonomy in implementation and being prescriptive with the evidence-based curriculum. Interventions that have improved emotion regulation skills in adolescents have shown the necessity of practitioners possessing a mix of complex therapeutic skills, the ability to form relationships with adolescents and the ability to internalise intervention concepts (Skeen et al., 2019; Pote, 2021). W4C attempts to develop these skills through appropriate training that incorporates local contextual understandings and curriculum-based activities, which enables coaches to provide a non-judgemental environment of emotional and physical safety as well as teach self-regulation skills.

Adolescents trusted the coaches and felt that they were easier to confide in than therapists or social workers. The preference for non-specialist interventionists has been expressed by participants in other studies (Padmanathan, 2013; Magidson et al., 2019). This type of relationship with peer mentors has been found to lead to positive impacts on mental health outcomes in other SfD and surf therapy interventions (Holt et al., 2017; Marshall et al., 2019). The presence of a consistent, supportive and caring adult who understands the context and life stage may be able to prevent or reverse the effects of a toxic stress response to adversity (Centre on the Developing Child, 2014), demonstrating the appropriateness of this approach.

There were some concerns from psychologists regarding the coaches' capacity to work with youth, having faced similar traumas themselves. To address this, the organisation ensures that there is a clear internal referral pathway for adolescents and provides regular counselling, individual and group supervision and a supportive organisational culture for the coaches, which was seen as acceptable and feasible by the coaches. W4C is particularly supportive in providing psychological therapy opportunities for their coaches, whereas most task-shared programmes in LMICs are not able to do so due to resource constraints. In most contexts, supervision has to play the role of tutorage as well as emotional support for peers dealing with challenging issues (Atif et al., 2017).

Concerns raised by a few children regarding bullying by other adolescents or unfair reprimands also highlight the importance of group management capacities by youth coaches. Issues raised by the adolescents in the FGDs were reported anonymously to the organisation to address with the relevant coaches.

The organisation uses catchy and adolescent-appropriate phrases and gestures, such as the 'Take 5' and having a 'banana culture' to reinforce learnings. This use of culturally appropriate terms and colloquial expressions improves the appropriateness and adoption of skills (Verhey et al., 2020). Adolescents also demonstrated satisfaction with the self-regulation tools they were learning, which is important as the acquisition of these skills has been found to decrease adolescent involvement in high-risk behaviours (Petitpas et al., 1999). In addition, the delivery of the intervention in a group format allowed adolescents to have fun, build social skills, feel a sense of belonging, participate in teamwork and learn from their peers. Group interventions have been endorsed by other studies as an acceptable and appropriate way to combat isolation, make new friends, create a sense of community and receive peer support (Ceccarelli et al., 2023).

The organisation's approach to training and supervision was crucial to all implementation outcomes. The training is intentionally used as a foundation to teach coaches how to create a safe space, be a caring and trustworthy adult and provide enthusiasm and energy, through which self-regulation can then occur. The intentionality of creating safe spaces was demonstrated by W4C coaches, which is important given that this is not necessarily inherent within other SfD initiatives (Marshall et al., 2024).

This underlies the importance of training in foundational concepts in any psychosocial intervention before any curriculum items or skills are taught. Notably, all stakeholders focused their comments on coach implementation and adoption of factors such as creating safe spaces and being caring adults and spoke less about actual curriculum items. This approach can be seen as a 'sector disruptor' from the more typical task-shared approach, where there is often minimal training due to financial constraints and an expectation of reliance on and understanding of manuals by peer-counsellors (Davies et al., 2022).

On-the-job group training and supervision throughout the year provided a platform for coaches to learn and practice activities included in the curriculum every week, ensuring the adoption, fidelity and feasibility of then teaching these to the adolescents. The approach of experiential learning and emotional support for coaches has been found to improve outcomes in other studies (Atif et al., 2017; Rose et al., 2025) and encourages positive feedback loops of learning, peer mentor feedback and input and maintaining motivation among mentors (Chibanda, 2017). This approach is then similarly used in teaching adolescents through integrating sports, role-plays, repetition and catch phrases and hand gestures, which improves their ability to generalise these skills to other situations in their lives (Petitpas et al., 1999).

Beyond the post-programme surf-club, the provision of other opportunities (fifth Pillar) was infrequently described by stakeholders, which may be due to the absence of opportunities outside of the programme or a lack of perceived relevance of this Pillar to the overall modality of teaching self-regulation skills to deal with everyday challenges.

Staff members identified that it is challenging to measure fidelity to the Method, performance of coaches and 'soft' outcomes for beneficiaries. In Zimbabwe, Chibanda et al. (2017) also discussed the difficulty of measuring fidelity to a task-shared intervention. Open-ended interviews and focus groups, as used in the current study, are useful to assess emotional outcomes from beneficiaries but they are also time and resource consuming.

Although a few adolescents expressed safety concerns about the ocean, activities such as surfing, swimming, and spending time on the beach were key to acceptability and programme attendance. These have specific tacit benefits, such as being in a 'blue space' (Britton et al., 2020), natural sensory grounding, respite (Marshall et al., 2023) and learning appropriate risk-taking (Spaaij and Schulenkorf, 2014). More explicit benefits include that it engages adolescents in the programme through being a fun and novel activity that most adolescents do not get the opportunity to participate in due to their socio-economic circumstances. It also allows adolescents to have fun in a non-competitive environment while challenging their fears and feeling a sense of achievement and mastery (Marshall et al., 2019).

While surfing is an acceptable vehicle for the 5-Pillar Method, the literature suggests that many of the core mechanisms of change in SfD initiatives are not inherently surfing-related (Holt et al., 2017; Boelens et al., 2022; Marshall et al., 2024) and that the five pillars can be transferrable across a wide variety of sports. Therefore, to improve the penetration and sustainability of the Method, W4C is expanding its training to other sporting interventions with adolescents, in partnership with the Department of Cultural Affairs and Sport, thus enhancing the feasibility for transference and penetration to non-coastal settings. These sporting interventions will, in turn, need to include elements found to improve mental health in organised sports activities that W4C includes, such as having "safe and appropriate peer interactions, structure and adult supervision, forming of supportive relationships with peers and adults, a sense of belonging and support of efficacy, mattering and skill-building" (Mahoney et al., 2005, 438).

### Recommendations for research, policy and practice

This study highlighted several factors relevant to the effective implementation of other SfD programmes for adolescents in LMICs. The findings emphasise the importance of ongoing training and supervision of youth coaches throughout the programme to support the adoption, fidelity and feasibility of competently teaching skills to adolescents. Employing young coaches from the same community as the adolescents contributes to the programme's relevance, as these coaches bring personal experience and a deeper understanding of the adolescents' contextual challenges (Holt et al., 2017). Role modelling of behaviours—by organisational staff for coaches and by coaches for adolescents—also plays a key role in shaping an appropriate and acceptable learning environment (Hurd et al., 2009). Additionally, offering psychological support for coaches and establishing clear referral pathways for traumatic disclosures are essential for ensuring programme feasibility and sustainability. Providing transport and meals supports better attendance and improves the acceptability of the programme. The year-long duration of the intervention was found to be feasible, and a follow-up weekend surf club created opportunities for ongoing engagement. Using simple, culturally relevant terminology and repeating activities helps to ensure appropriateness and reinforces learning among adolescents. Including skills training for parents could further extend the impact of such interventions (Ceccarelli et al., 2023). Finally, applying a trauma-informed and context-sensitive approach to mental health support is vital to meet the needs of adolescents living in poverty and adversity. Creating a fun, non-competitive environment further enhances the appropriateness and acceptability of such interventions (Petitpas et al., 1999).

Given the findings of the current study, future research should consider exploring the perspectives of community leaders, mental health providers and policymakers in relation to the provision of SfD programmes for adolescents in different contexts. Another area of research could involve exploring the competencies of coaches, a more detailed assessment of whether they deliver the service with fidelity, as well as techniques for the scale-up of supervisory support and psychological support in interventions with few resources and psychological specialists. Qualitative research with adolescents who have completed the full year-long programme will also be insightful to explore impact. Further targeted research is recommended to better understand the dosage and behaviour change. The study serves as a base from which to conduct an effectiveness randomised controlled trial of the W4C intervention using the 5-Pillar Method (Seward et al., 2021).

### Limitations

The study had several limitations. Fidelity to the manual was not assessed in detail by independent raters; instead, it relied on self-reports from coaches and reflective reports from other participants. Member checking with participants was not conducted, which could have strengthened the validity of the findings. The evaluation did not include cost as an outcome due to the absence of a suitable costing tool and related data. Some of the coaches who participated in the focus groups had only been at the organisation for 4 months at the time of the interviews. However, this was intentional to ensure a diverse sample, as other coaches had been involved for over a year. Adolescents were interviewed 6 months into the intervention and, therefore, had not completed the full year-long programme. The study also lacked data on the characteristics of adolescents who attended versus those who did not, limiting the ability to assess the intervention's acceptability across different backgrounds. Additionally, the research was conducted in a South African context, and caution should be exercised in generalising the findings to other settings. Finally, due to the qualitative nature of the study, the possibility of social desirability bias among participants cannot be excluded.

### Conclusion

This study provides qualitative insights into the implementation outcomes of the 5-Pillar Method used by the W4C programme in South Africa. Over the past 13 years, the programme has achieved significant penetration into local communities, growing from 15 to over 1,500 annual participants, while maintaining strong partnerships with schools and healthcare providers. A prevention approach using a trauma-sensitive evidence-based curriculum was seen to be appropriately responsive to the high levels of adversity and trauma that adolescents experience. Stakeholders found the process of using safe physical and emotional spaces, fun, group-based ocean activities, caring youth coaches from the same communities, and modelling and repetition of skills to be acceptable, notwithstanding some concerns regarding coaches' psychological capacites. Implementation of the Method made feasible through consistent on-the-job training and supervision in the creation of safe spaces and curriculum items and the provision of mentoring and counselling for coaches. This facilitated an optimal environment for adolescents to learn social and emotional regulation skills, which may act as protective factors against poverty-related adversity and toxic stress. It is hoped that this will prevent future mental illness and reduce the burden of adolescent ill-health. These findings underscore the transformative potential of combining sport-based interventions with psychosocial task-sharing to build mental health resilience in at-risk adolescent populations.

**Open peer review.** To view the open peer review materials for this article, please visit http://doi.org/10.1017/gmh.2025.10033.

### Abbreviations

| | |
|---|---|
| **LMICs** | Low- and middle-income countries |
| **NGO** | Non-governmental organisation |
| **SfD** | Sport for Development |
| **W4C** | Waves for Change |
| **WHO** | World Health Organisation |

**Supplementary material.** The supplementary material for this article can be found at http://doi.org/10.1017/gmh.2025.10033.

**Data availability statement.** The data that support the findings of this study are available upon request from the corresponding author, TD. The data are not publicly available due to the sensitivity of the research, the involvement of vulnerable adolescent participants and naming of the organisation involved, and thus potential ability to identify adult participants.

**Acknowledgements.** The authors would like to thank all the participants for giving their time to participate in this study, and particularly the staff from Waves for Change who provided background information, data and clarifications.

**Author contribution.** Research conception and design: TD and CL. Data acquisition: TD. Analysis, interpretation of data, drafting and revising of the work: TD, CL and JM. Contextualisation of data: NvD, PY and TC. Approval of the submitted version: All authors. The authors agree to be personally accountable for their own contributions and to ensure that questions related to the accuracy or integrity of any part of the work are appropriately investigated and resolved.

**Financial support.** This work was partially funded by Grand Challenges Canada.

**Competing interest.** N van der Merwe and P Yarrow are employed at Waves for Change, and T Conibear is the founder and CEO of the organisation.

**Ethical approval.** The research was performed in accordance with the Declaration of Helsinki. Ethical approval for the study was obtained from the University of Cape Town Health Sciences Faculty Human Research Ethics Committee (HREC), reference number: 790/2022. *Consent to Participate:* Written informed consent was obtained from all participants and the caregivers of the adolescents, together with assent from the adolescents.

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
