## [Reviewer Report]

Dear Authors,

Thank you for the opportunity to review your manuscript, “Implementation outcomes of the Waves for Change 5-Pillar Method for community-based task-shared prevention and promotion of adolescent mental health in South Africa.” Your work addresses an important area of research, and I appreciate the effort that has gone into conducting and reporting on this study. Below are my comments and suggestions for improvement:

Introduction:

• The framing of why assessing implementation outcomes is relevant could be more explicit. It would help to highlight what this type of study contributes compared to studies that solely lay in the research space.

Methods:

• It would be useful to expand the setting section to also add information on the years in which the intervention was implemented

Results:

• General

o Starting with the report of penetration feels not exactly intuitive. Consider moving around the paragraph of the results to something along the lines of: appropriateness, adoption, acceptability, feasibility, fidelity, penetration, sustainability (in line for example with the progression highlighted by Wgenaar and colleagues 10.1017/gmh.2020.1) , which could help the reader reflect more easily on the implementation process

o The Proctor model includes costs as an outcome; it would be useful to acknowledge this as a limitation and explain why cost data was not reported

• Penetration: this section would benefit from additional information on the context, such as the number of schools in the area or the total number of potentially eligible students, etc. that would make it more closely fit with the Proctor definition. Currently, the findings seem to reflect more reach than penetration

• Feasibility: I would suggest moving the attendance data into this section, as it seems to align more with feasibility than with the other outcomes. Some of the quantitative sources referenced in other sections might fit better here.

• Fidelity: Attendance data may be more relevant to feasibility than fidelity, which is generally concerned with adherence to the intervention’s intended design. Table 4 and related reflections on fidelity remain relevant but might benefit from further clarification.

Line specific comments

• Line 311: specify who perceived the intervention as appropriate—participants?

• Line 519: consider referring also to Rose et al. (10.1080/15374416.2022.2151450) for more explicit work on task-sharing.

• Line 563: I would suggest to clarify how Ceccarelli et al. support the claim that training and open space vs. reliance on curriculum is beneficial.

• Line 599: the discussion on feasibility could benefit from addressing the role of training and supervision, which the literature, but also the findings of the work, highlight as key factors in successful implementation

---

## [Reviewer Report]

Thank you for inviting me to review this paper describing the Waves for Change intervention. The study aimed to assess the implementation outcomes of W4C’s 5-Pillar Method. The programme sounds fantastic, innovative and important. It was interesting to read in detail about the activities and its success.

That being said, I have reservations about the paper in its current form. However, my comments should not deter the authors from sharing their excellent work and disseminating the programme, and I hope will serve to strengthen their analyses and the usefulness of the paper for the wider Global Mental Health field.

Main comments

Whilst the paper does a good job of describing the intervention in detail the analysis does not appear to represent a balanced view of the challenges and opportunities or strengths and weaknesses of the intervention and lacks a critical angle. The findings are almost without exception positive with regards to the success of the intervention, without consideration of deviant cases or a critical perspective. The description of the analysis hints at possible bias: using a deductive coding approach researchers “searched for evidence of” implementation, but incorporation of inductive codes enabling themes of potential harms or barriers to implementation to emerge from the data would have strengthened the analysis and made for a more informative paper. As such, the paper is at risk of reading as a promotional report rather than a research paper. Whilst I note that the lead and senior authors are external to the Waves for Change team, two of the authors are employed by Waves for Change and one is the CEO and founder. These authors were consulted about the analysis (line 233) to “ensure reliability and validity”. It is not clear how they were able to remain impartial to the study findings.

I also have concerns about the quality of the qualitative and quantitative data analyses. The data collection procedure lacks important detail. For example, the sampling approach for participants was random number generation but this is not justified in the text. Arguably it would have been more informative to sample purposively, ensuring representation across e.g. gender, age group, attendance level. (N.B. Demographic characteristics of the participants are not provided so it is unclear if the sample was representative). There is no description of the topic guides and how these were developed. What language were the interviews/FGDs conducted in? Were they transcribed and if so by whom and verbatim? What was the duration of interviews and FGDs. It is also unclear how social desirability bias was mitigated in these interviews/FGDs. Did the coaches and participants feel free to speak about what wasn’t working and what needed to change about the intervention? How did researchers ensure this? Why did the researchers only conduct FGDs with adolescents and not interviews? Would adolescents feel able/comfortable to share their thoughts about the programme and their experience in front of their peers? There are also missed opportunities to validate the findings, for example through a member check with the participants or an alternative group of Wave for Change coaches/adolescents. The authors might consider using the COREQ Checklist to ensure all of these details are reported in the paper.

Regarding the quantitative analysis, more detail would also be helpful. For example, how were the historical documents and annual reports analysed? What were the routine site assessments, what did they assess and how were they analysed? Were there any quality checks (e.g. accuracy of data entry, inter-rater reliability) in place? It is also hard to understand the relevance of the figures reported in the findings section without further information. For example Line 443, providing the number of adolescents at each session (25-92) does not give a sense of how many did not attend, nor of attendance rates and drop out over the programme duration. It would also be helpful to describe the characteristics of adolescents who did attend to assess whether the programme was acceptable to adolescents from diverse backgrounds or only from a specific subgroup. Moreover, it would be helpful to have more information about the site assessment (line 458), including the percentage of sessions assessed.

I also have more specific comments:

• The Waves for Change intervention is conceptualised as a prevention and promotion intervention in the paper but I’m wondering if a programme targeting at risk adolescents can be considered as a promotion intervention.

• The authors must be careful not to overinterpret the findings. For example on lines 58-59 the authors suggest the findings support scale up of the programme in LMIC contexts. However, the study has not assessed the effectiveness of the programme which is a crucial piece of evidence in deciding whether a programme should be scaled.

Introduction

• Consider justifying the inclusion of the study by Beranbaum et al as a sample of 83 adolescents is very small in terms of inferring population level trends.

• More information about how sport could help to reduce the treatment gap would strengthen and focus the introduction.

Methods

• Line 171 – “and to how make”

• Line 173 “coaches then then”

• Line 253 and 254– data are plural

• Line 253 and 254– data are plural

Results

• An overview of the demographic characteristics of participants, especially adolescents, is needed.

• Line 289 - Did the participants really say that “having an activated nervous system” was a challenge they faced? Was this a direct quote? Better to use their own language to demonstrate closeness to the data.

• Line 305 – the authors mention “their original child-led study” without explaining the relevance and findings from the study.

• More specificity in terms of who said what in reporting the analysis would demonstrate sensitivity to the data. For example, line 301 – “The organisation’s approach to teaching and learning social and emotional skills through creating safe physical and emotional spaces, and fun tasks using sport, roleplays, and repetition, was seen as compatible and appropriate for adolescents, to aid delivery and enable adoption of the skills into their day to day lives.” Who said this? The adolescents? The parents? The coaches? Also, lines 310 and 311 – “were seen as appropriate and relevant and suitable” by whom? In what ways?

• Lines 320-321 – “The organisation has a clearly structured referral system” – Did the participants describe this referral system? How was it clearly structured? What worked about the referral system and what didn’t?

• Line 332 - “Through their youth and enthusiasm, the coaches’ ability to engender a relaxed, enthusiastic, and non-judgemental environment for adolescents creates an emotionally safe space, which demonstrates credibility of the 5-Pillar Method.” This statement needs to be backed up with evidence from the data. How did coaches create the environment? Can you give examples from the data that show how adolescents perceived the environment as emotionally safe?

• Line 369 – “the success of the Method was seen to be maintained through consistent support for coaches… “ what is the evidence for this? Which participant group said this? Did different groups value different components more than others? Comparing and contrasting perceptions across groups could deepen the analysis.

• Line 377 – Did all adolescents say the year long programme and frequency of sessions was feasible and acceptable, or did any suggest alternatives? I would be surprised if all agreed without exception and wonder if the environment of the FGD enabled them to share freely.

• Line 476 – It would be informative to clarify where the external funding is from, e.g. if from the government or private donors?

Discussion

• Lines 510-513 – I do not think that the analysis presented in the paper fully supports this. Were adolescents coming for the skills regulation and trauma responsiveness or because they enjoyed surfing?

• It would help to strengthen the discussion if the authors could comment on the relevance of the surfing component. How important was this for the success of the programme? How acceptable was surfing among different adolescent subgroups? Could the programme consider incorporation of other sports to appeal to a wider group of adolescents, or to enhance its feasibility in non-coastal settings?

• Line 608 – I do not think there is evidence in the paper to assert that “a year-long programme was deemed appropriate and suitable to entrench learnings and behaviour change”.

---

## [Reviewer Report]

The manuscript describes the implementation outcomes of prevention and promotion interventions delivered through a task-sharing approach in the Western Cape, South Africa, using a mixed-methods design, following the Proctor framework. I read the manuscript with great interest and have only minor comments to enhance the relevance of the intervention and its public health impact. These strategies are rarely evaluated through implementation outcomes, making this assessment particularly valuable in determining their real-world applicability.

- Line 160: I would refer to the “participants” as “adolescents” to distinguish them from the actual study participants, who include all stakeholders.

- You name the intervention as promotion and universal prevention based on its content rather than the target population itself, correct? Because if it were based on the population, it might fall under selective prevention (for “at-risk adolescents”). It’s not a major issue, but I would clarify the reasoning so that public health readers can align with the labeling.

---

## [Reviewer Report]

An important investigation into this method of delivering sport for development. I found this to be a well-constructed mixed methods study that gave context to the study, acknowledged its limitations and potential conflicts, and appropriately attempted to mitigate these. I found the manuscript easy to follow and comprehensive. The practical recommendations were a good summary of learnings from the study. From my perspective, it can be published in its current form.

In the proofing stage: Line 420 - italics should be edited to only include the quote.

In the proofing stage: Line 420 - italics should be edited to only include the quote.

---

## [Editor Report]

Thank you for submitting your article on ‘Implementation outcomes of the Waves for Change 5-Pillar Method for community-based task-shared prevention and promotion of adolescent mental health in South Africa’. Please would you attend to the reviewers' comments and re-submit. Please pay special attention to comments made by reviewer 1, who requests major revisions and reviewer 2 who requests minor revisions.

---

## [Reviewer Report]

Dear Authors, thank you for your thorough revisions and thoughtful responses. I’m satisfied with the changes and have no further feedback.